# Leveraging Temporal Graph Networks Using Module Decoupling

## Abstract

Modern approaches for learning on dynamic graphs have adopted the use of batches instead of applying updates one by one. The use of batches allows these techniques to become helpful in streaming scenarios where updates to graphs are received at extreme speeds. Using batches, however, forces the models to update infrequently, which results in the degradation of their performance. In this work, we suggest a decoupling strategy that enables the models to update frequently while using batches. By decoupling the core modules of temporal graph networks and implementing them using a minimal number of learnable parameters, we have developed the Lightweight Decoupled Temporal Graph Network (LDTGN), an exceptionally efficient model for learning on dynamic graphs. LDTG was validated on various dynamic graph benchmarks, providing comparable or state-of-the-art results with significantly higher throughput than previous art. Notably, our method outperforms previous approaches by more than 20% on benchmarks that require rapid model update rates, such as USLegis or UNTrade. The code to reproduce our experiments is available at this http url.

## 1 Introduction

Dynamic graphs are commonly used to describe real-world dynamic systems, where the interacting elements are modeled as nodes, and the interactions between two elements are represented as edges. Each edge is usually labeled with a timestamp indicating its time of occurrence. Item recommendation on e-commerce platforms (Ding et al., 2019), friendship suggestion on social networks (Backstrom & Leskovec, 2011; Haghani & Keyvanpour, 2019), anomaly detection on communication networks (Yu et al., 2018), and traffic forecasting (Cini et al., 2023) are all practical tasks that can be modeled using dynamic graphs.

Although most graph-related real-world tasks are time-evolving, deep learning approaches usually focus on problems described using static graphs. Moreover, it had also been shown that ignoring the dynamic nature of a system by abstracting it with static graphs is inadequate (Rossi et al., 2020; Xu et al., 2020). A dynamic representation of a system, on the other hand, is often able to define the evolving behavior of the latter (Simmel, 1950; Granovetter, 1973; Mangan & Alon, 2003; Toivonen et al., 2007; Gorochowski et al., 2018).

Dynamic graph approaches are often based on either discrete-time (Liben-Nowell & Kleinberg, 2003; Sankar et al., 2020; Pareja et al., 2020) or continuous-time (Trivedi et al., 2019; Ma et al., 2020; Cong et al., 2023) settings. In discrete-time settings, data are received as a sequence of snapshots describing the full graph structure at specific times, while in the flexible continuous-time setting, a single update on the graphs can happen at any moment. The setting in which the deep learning models for dynamic graphs operate at inference time can be roughly divided into the following types: streaming, deployed, and live-update (Huang et al., 2023). In this work, we focus on continuous-time dynamic graphs in the context of streaming in which the models may update upon receiving new information, but they usually cannot perform backpropagation due to the high throughput required.

In the streaming setting for continuous-time dynamic graphs, deep networks usually have to use batches to keep up with the stream of incoming updates, which means they process multiple updates in parallel. This situation introduces a new problem of *missing updates* – updates for the models that are not being considered for the predictions of inputs inside their common batch. In this work, we

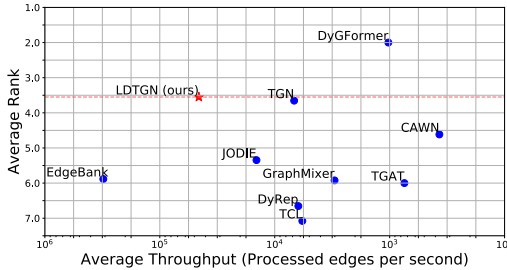 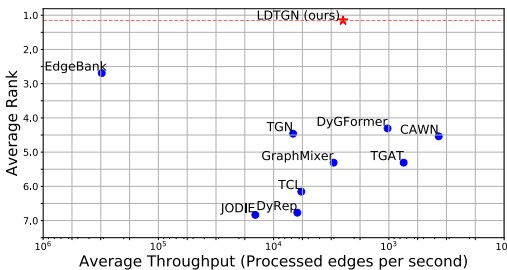

(a) Comparison with baseline models. The average throughput and average rank were computed over all the future edge prediction benchmarks described in Section 5.1.

(b) Comparison with baseline models. The average throughput and average rank were computed over all the online future edge prediction benchmarks described in Section 5.2.

Figure 1: Comparison with baseline models.

suggest a strategy to minimize the frequency of *missing updates* while still using batches. Guided by this strategy, we have built the Lightweight Decoupled Temporal Graph Network (LDTGN) – an efficient model for dynamic graph learning that outperforms the vast majority of many other well-known baselines, both in terms of running time and performance, as illustrated in Fig. 1a.

As noted above, in the streaming setting, models usually cannot perform backpropagation at inference time since this operation is excessively expensive. In Section 5.2, we demonstrate the importance of using backpropagation at inference time in addition to solely applying updates in the scenario of online future edge prediction, as depicted in Fig. 1b. Moreover, we show that when using backpropagation with LDTGN, the model keeps the throughput within the range of other standard baselines, making it suitable for the online learning scenario with high streaming rates.

To summarize, this work makes the following contributions:

- We suggest a novel methodology for building deep learning models for dynamic graph tasks.

- Based on the suggested methodology, we propose a new lightweight model for dynamic graph learning tasks that can operate at high streaming rates.

- Based on our experiments, our lightweight model outperforms many baselines and achieves state-of-the-art performance on various dynamic graph benchmarks.

## 2 BACKGROUND

Static graph $\mathcal{G} = (\mathcal{V}, \mathcal{E})$ is a tuple of vertex set $\mathcal{V}$ and edge set $\mathcal{E}$, s.t., $e \in \mathcal{E}$ is a tuple of two vertices from $\mathcal{V}$. $\mathcal{G}$ is often equipped with a features function $F_\mathcal{V} : \mathcal{V} \to \mathbb{R}^n$ or $F_\mathcal{E} : \mathcal{E} \to \mathbb{R}^n$ that maps a vertex or an edge into an n-dimensional vector representing their matching features. Continuous-Time Dynamic Graph (CTDG) is a sequence $\mathcal{Q} = \{x_{t_1}, x_{t_2}, ...\}$ of timestamped updates on the graph. An update $x_t$ that occurs at time $t$ can be one of the following: node addition, node removal, edge addition, and edge removal. Each update may include an n-dimensional vector describing its features. A *snapshot* of $\mathcal{Q}$ at time $t$ is the static graph received by applying all the actions that have occurred until time $t$.

Deep learning models designed to learn on CTDG usually process the sequence of updates by using batches to achieve reasonable throughput during inference time (Kumar et al., 2019; Rossi et al., 2020; Wang et al., 2021b; Cong et al., 2023). In the streaming setting, where the graph receives new updates at extremely high speeds, it is crucial for the model to have sufficient throughput. Otherwise, a buffer to the model containing the new updates will overflow.

Models for dynamic graph learning tasks often suffer from the undesirable phenomenon of *missing updates* when using batches. This phenomenon occurs when a batch contains crucial updates to make correct predictions for subsequent inputs in the same batch. In such scenarios, the model will give predictions based on its current state, which does not include the updates in the batch. *Missing*

*updates* are less likely to occur when using smaller batches since, in these circumstances, the models update more frequently. Therefore, in the streaming setting, there is a trade-off with respect to the batch size.

As a result of the growing interest in the CTDG scenario with a stream of updates, several techniques were recently developed (Kumar et al., 2019; Trivedi et al., 2019; Xu et al., 2020; Wang et al., 2021a;b; Cong et al., 2023). Many of these methods are specific cases of the Temporal Graph Network (TGN, Rossi et al., 2020) model. TGN is a general deep learning architecture designed to learn on CTDG while achieving throughput suitable for streaming tasks. TGN can be said to have two central modules: memory and prediction.

**Memory module**   The memory module is responsible for updating the state of the nodes in the graph as seen by the model. When a new batch of updates arrives, the memory module applies a message function that generates a message vector for each node involved in each update. If the update is an interaction $e_{i,j}$ between nodes $v_i$ and $v_j$ at time $t$, their appropriate messages are:

$$m_i(t) = msg_s(s_i(t^-), s_j(t^-), \Delta t, e_{i,j}), m_j(t) = msg_d(s_j(t^-), s_i(t^-), \Delta t, e_{i,j}) \qquad (1)$$

where $s_i(t^-)$ is the state of $v_i$ prior to $t$. $msg_s$ and $msg_d$ may have learnable parameters. Then, all the messages in the batch are aggregated into a single message per node:

$$\overline{m}_i(t) = agg(m_i(t_1), m_i(t_1)...m_i(t_n)) \qquad (2)$$

Here $t_1 \leq t_2 \leq .. \leq t_n = t$. The aggregation function, for example, can take only $m_i(t_n)$ and neglects any previous messages in the batch. Finally, the message updater updates the state of the model:

$$s_i(t) = mem(\overline{m}_i(t), s_i(t^-)) \qquad (3)$$

The $mem$ function usually contains learnable memory such as LSTM (Hochreiter & Schmidhuber, 1997) or GRU (Cho et al., 2014).

**Prediction module**   The prediction module gives the predictions for the inputs in a batch. First, it reads from the memory module all the states of the nodes in the neighborhood of any input node, i.e., the source node and destination node in an interaction input, and their appropriate neighbors. Then it generates a new embedding for each node in the input based on its neighborhood state. For node $v_i$ its embedding formulation is:

$$z_i(t) = \Sigma_{j \in n_i} h(s_i(t^-), s_j(t^-), e_{i,j}, v_i, v_j) \qquad (4)$$

where $n_i$ is the neighborhood of $v_i$, and $h$ is a learnable function. Note that the prediction for time $t$ is based on the state prior to $t$. Using the neighborhood of a node in the graph to compute its embedding averts the staleness problem (Kazemi et al., 2020).

As mentioned above, most standard approaches for learning on CTDG suffer from *missing updates*. A good example of this is TGN with an attention aggregation on the Canadian Parliament benchmark for future edge prediction (Poursafaei et al., 2022).

The Canadian Parliament is a dataset describing a dynamic political graph, where the nodes are the members of the Canadian Parliament and there is an edge between two members if they both vote yes on a bill. In practice, for the task of edge prediction for this dataset, it is relatively simple to predict whether a given temporal edge at a specific time is real and positive or negative and randomly sampled. This is because the dataset is built of sequences of edge updates (multiple edge updates per sequence) and in each sequence, the source nodes of the edge updates are the same. When using a standard batch size of 200, TGN usually misses the switch from one sequence to another as illustrated in Fig. 2b, and, therefore, frequently gives wrong predictions.

In Fig. 2a we present the performance of TGN as a function of its batch size on the Canadian Parliament dataset. We can clearly see that large batches prevent TGN from maximizing its potential, even for a relatively simple benchmark.

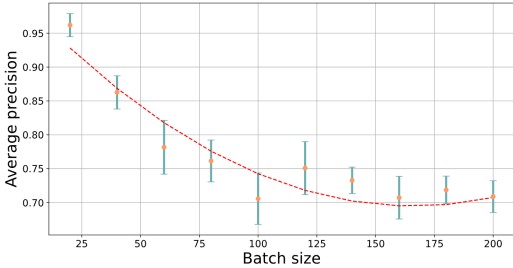 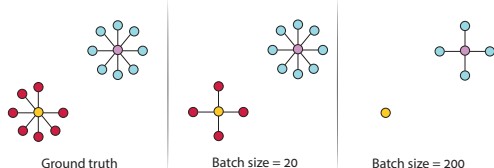

(a) Average precision of TGN with attention aggregation on the Canadian Parliament benchmark for future edge prediction using different batch sizes. The means and standard deviations over five different runs are reported as circles and ranges, respectively.

(b) Simplified illustration of the Canadian parliament graph as seen by TGN with batch sizes of 20 and 200, at the same moment in time. With a batch size of 200, TGN misses the change of sequence of updates from updates to the purple node to updates to the yellow node, while with a batch size of 20, TGN manages to catch the change and a few updates to the yellow node.

Figure 2: TGN performance on the Canadian Parliament dataset.

## 3 RELATED WORK

**Handling *missing updates*** The t-Batch algorithm (Kumar et al., 2019) was originally intended to improve the running-time performance of deep networks for dynamic graphs that process one update after the other (i.e., batch size equal to 1). The logic motivating t-Batch is that these networks can combine multiple updates into a single batch and apply them in parallel if they do not contain the same nodes, where the batches are temporally sorted. Using t-Batch, the model JODIE becomes X9.2 faster than similar methods without suffering from *missing updates* (Kumar et al., 2019). The t-Batch algorithm, however, suffers from two main flaws. First, large batch sizes for t-Batch are often impossible since temporal locality is a common characteristic of dynamic graphs (Poursafaei et al., 2022). In addition, many modern deep learning networks for dynamic graphs, such as TGN, depend on the neighborhood of the nodes to give an appropriate prediction, causing t-Batch to perform complicated neighborhood-independent batches instead of node-independent batches, which are significantly smaller.

**Efficient methods for streaming** According to Huang et al. (2023), EdgeBank is currently an order of magnitude faster than other well-known techniques for dynamic graphs. EdgeBank (Poursafaei et al., 2022) is a memorization algorithm that saves any seen update and predicts according to a simple decision rule that can be one of the following: whether the input was seen in the last few iterations or whether the input has already been seen a sufficient number of times. The algorithm's simplicity allows it to perform extremely fast, even without using batches at all, thus not suffering from *missing updates*. Nevertheless, EdgeBank was developed to serve as a baseline for testing and comparing other methods for dynamic graphs (Poursafaei et al., 2022), and, therefore, its performance lags significantly behind the state-of-the-art (Yu et al., 2023; Huang et al., 2023).

## 4 METHOD

In the following section, we describe a method to balance the batch size trade-off described earlier in Section 2. To do this, we **decouple** the TGN modules. Each module uses a different batch size. In general, the memory module will use smaller batch sizes for frequent updates and the prediction module will use larger batch sizes for efficiency.

Following that, we describe our proposed lightweight model for dynamic graphs. The model is a TGN with decoupled modules that are implemented using efficient functions. Specifically, we parameterize the EdgeBank model to allow it to learn. Then we add extra parameters to consider single-node information in the prediction instead of solely relying on edge information.

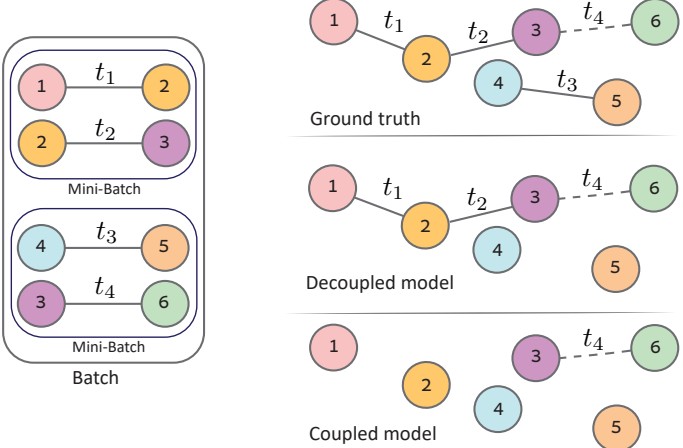

Figure 3: Coupled and decoupled models states compared to the ground truth at $t_4^-$ for predicting the edge $(v_3, v_6)$. The coupled model was previously updated at $t_1^-$ and, therefore, does not contain $(v_1, v_2)$, $(v_2, v_3)$ and $(v_4, v_5)$. The decoupled model was previously updated at $t_3^-$ and, therefore, does not contain $(v_4, v_5)$.

## 4.1 THE DECOUPLING STRATEGY

To this end, we propose to increase the **decoupling** between the core modules of TGN: the prediction module and the memory module. We do this by first saving the neighborhood state in addition to the node state:

$$S_{n_i}(t) = \{(s_j(t), e_{i,j}, v_j) \,|\, j \in n_i\} \tag{5}$$

Next, given a batch of updates to apply and inputs to predict, the batch is divided by the model into small consecutive batches called mini-batches. The memory module operates on the mini-batches and thus it is able to perform memory updates more frequently as demonstrated in Fig. 3. After processing a mini-batch, but before processing the next one, the memory module extracts the node states and the neighborhood states relevant to the next mini-batch to prevent their override. Note that extracting the states is prior to the memory update and consequently *missing updates* may still occur. In Fig. 3 we have a *missing update* in the first mini-batch for the prediction of the interaction between $v_2$ and $v_3$ since at this time, the decoupled model is unaware of the new interaction between $v_1$ and $v_2$. The prediction module operates only after the full batch is processed by the memory module. It uses the states extracted by the memory module as input and gives the appropriate predictions. In Fig. 3 we managed to prevent a *missing update* for the interaction between $v_3$ and $v_6$ by using the decoupling strategy since the memory module is aware of the interaction between $v_2$ and $v_3$.

**Decoupling** the memory module and the prediction module offers two immediate benefits. First, by decoupling the memory module from the prediction module and setting the mini-batch size to 1, we completely solve the *missing updates* problem. Secondly, we can accelerate the running time of an existing model while keeping its accuracy the same by decoupling its modules and setting the mini-batch size to be equal to the model's original batch size, and then increasing the new batch size of the prediction module significantly. Using the same batch size for the mini-batches ensures the same frequency of missing updates and the new larger batch size will improve the performance in terms of throughput.

## 4.2 LIGHTWEIGHT DECOUPLED TEMPORAL GRAPH NETWORK

The EdgeBank model can be formulated as a memory-based algorithm as presented by Poursafaei et al. (2022), but we can also describe it as a linear function that maps a time-based difference into a prediction. Eq. (6) describes the linear function of EdgeBank with a decision function that considers any edge $e_{i,j}$ that appeared in the last 1000 updates as positive.

$$p = -(t - t_{i,j}) + 1000 \tag{6}$$

We can parameterize the threshold of 1000 suggested by Poursafaei et al. (2022) and receive Eq. (7).

$$p = (t - t_{i,j})w + b \tag{7}$$

Using Eq. (7) we can learn the right threshold for each task. As in EdgeBank, this function does not incorporate the nodes themselves into the prediction. This can easily be solved by adding the time distances of each node as in Eq. (8).

$$p = (t - t_{i,j})w_1 + (t - t_i)w_2 + (t - t_j)w_3 + b \tag{8}$$

Eq. (8) is the prediction function that will be used by our model.

We now describe the implementation details of LDTGN vis-à-vis each of its modules for the task of future edge prediction, using the decoupling strategy.

**Memory module**   Given a new interaction update $e_{i,j}$ at iteration $t$, LDTGN will compute the following messages:

$$m_i(t) = m_j(t) = m_{i,j}(t) = t \tag{9}$$

Since in our implementation we set the mini-batch size to one, the message aggregator submodule is not required. The memory updater is formulated by the following:

$$s_i(t) = (m_i - s_i(t^-)_2, m_i), s_j(t) = (m_j - s_j(t^-)_2, m_j), s_{i,j} = (m_{i,j} - s_{i,j}(t^-)_2, m_{i,j}) \tag{10}$$

Note that in contrast to the standard TGN memory module, we also generate messages for the edges and save states for them.

**Prediction module**   Given an interaction $e_{i,j}$ as input, the embedding performed by the prediction module will be computed by the following:

$$z_{i,j}(t) = (N(s_i(t^-)_1), N(s_j(t^-)_1), N(s_{i,j}(t^-)_1)) \tag{11}$$

where $N$ is a normalization function that can be either *static* or *dynamic*. The normalization is required to keep the values of $z_{i,j}(t)$ between 0 and 1. The static normalization is calculated by:

$$N_{static}(x) = \frac{\log(x)}{\log(C)} \tag{12}$$

where $C$ is a predefined constant. In our experiments, we set $C$ to be the maximum number of iterations among all the datasets. The dynamic normalization also allows the model to set a threshold that is relative to the current iteration instead of a static one, and it is computed by:

$$N_{dynamic}(x) = \frac{x}{t} \tag{13}$$

Finally, if the edge has at least one previously seen node, the module will give it the following prediction:

$$p = w^T z_{i,j}(t) + b \tag{14}$$

Here, $w$ is a weight vector and $b$ is a bias. If $e_{i,j}$ does not have any previously seen node, the module predicts -1, i.e., a negative edge.

### 4.3 Training

While LDTGN was developed to be trained for the future edge prediction task, it can easily be adjusted to solve other tasks such as node classification. Furthermore, since the model requires a relatively small number of calculations to make predictions and has only a few learnable parameters, it is suitable for online-learning scenarios with high streaming rates. That is, even with minimal or no training examples, the model performs its core training during inference time using backpropagation while maintaining high throughput.

We trained LDTGN as if it were in an online-learning scenario, i.e., by iterating on the training data once using batches and performing a few iterations (epochs) of forward and backward propagation on a newly received batch of inputs. For a fair comparison in the experiments, we did not perform backpropagation at inference time in the experiments with the standard setting of transductive and inductive learning. Further details regarding the model training are provided in Appendix C.

## 5 Experiments

In this section, we present the results of the extensive experiments that we conducted. All the experiments were performed for the task of future edge prediction with random negative edge sampling on the following datasets: Wikipedia, Reddit, Mooc, lastFM, Enron, SocialEvo, UCI, Flights, Can.Parl, USLegis, UNTrade, UNVote, Contacts, which were collected by Poursafaei et al. (2022). Additional information and statistics regarding the datasets can be found in Appendix A. We used nine popular methods as baselines for the task of future edge prediction: JODIE (Kumar et al., 2019), DyRep (Trivedi et al., 2019), TGAT (Xu et al., 2020), TGN (Rossi et al., 2020), CAWN (Wang et al., 2021b), EdgeBank (Poursafaei et al., 2022), TCL (Wang et al., 2021a), GraphMixer (Cong et al., 2023) and DyGFormer (Yu et al., 2023). Additional information regarding the baselines can be found in Appendix B. For convenience, we do not present in this section the results of DyRep and TCL. The full results can be found in Appendix D.

We adopted the approach that was used in many existing works and split the dataset into training, validation and test sets by performing a chronological split of 70%–15%–15%. We report the mean and standard deviation of the Average Precision (AP) on the test set.

### 5.1 Future edge prediction

In the first experiment, we tested transductive future edge prediction. The results are presented in Table 1. We also performed an experiment for the inductive future edge prediction setting, in which all the edges in the validation and test sets must contain nodes that have not been previously seen in the training set. The results for this experiment are reported in Table 2. For both transductive and inductive settings, the results for all the baselines except EdgeBank were taken from (Yu et al., 2023), and the experiments were performed using a standard batch size of 200 and ideal hyperparameter configurations. For fairness, we recreated the experiments for EdgeBank with a batch size of 1, since even with a small batch size, its throughput is extremely high. We also tuned the threshold value of the prediction function of EdgeBank to be between the standard 1000 and a new threshold of 100k.

We calculated the average number of learnable parameters required for each model to achieve its best performance and reported it in Fig. 5. We also measured the average throughput at inference time for each model, where the throughput is defined as the number of edges the model can process in a single second. The results are shown in Fig. 6.

From all the results presented above, we conclude that our model is the best alternative among the tested models when considering both running time and performance.

### 5.2 Online future edge prediction

We also tested our model and the baselines in the online future edge prediction setting. In this setting, the models receive a few data samples for training and validation and are then tested on the rest of the data. This experiment setting is crucial since in many real world scenarios, there are a few to zero training samples from which to learn. Moreover, in continual dynamic systems with a stream of updates that operate over extended periods, the behavior of the systems may vary significantly

| Dataset | JODIE | TGAT | TGN | CAWN | EdgeBank | GraphMixer | DyGFormer | LDTGN (ours) |
|---|---|---|---|---|---|---|---|---|
| wikipedia | 96.50±0.14 | 96.94±0.06 | 98.45±0.06 | 98.76±0.03 | 94.41±0.02 | 97.25±0.03 | **99.03±0.02** | 94.43±0.03 |
| reddit | 98.31±0.14 | 98.52±0.02 | 98.63±0.06 | 99.11±0.01 | 95.78±0.02 | 97.31±0.01 | **99.22±0.01** | 95.80±0.01 |
| mooc | 80.23±2.44 | 85.84±0.15 | 89.15±1.60 | 80.15±0.25 | 71.46±0.08 | 82.78±0.15 | 87.52±0.49 | **94.08±0.04** |
| lastFM | 70.85±2.13 | 73.42±0.21 | 77.07±3.97 | 86.99±0.06 | 86.13±0.03 | 75.61±0.24 | **93.00±0.12** | 91.28±0.03 |
| Enron | 84.77±0.30 | 71.12±0.97 | 86.53±1.11 | 89.56±0.09 | 91.02±0.21 | 82.25±0.16 | 92.47±0.12 | **95.67±0.06** |
| SocialEvo | 89.89±0.55 | 93.16±0.17 | 93.57±0.17 | 84.96±0.09 | 94.64±0.02 | 93.37±0.07 | **94.73±0.01** | 93.12±0.06 |
| UCI | 89.43±1.09 | 79.63±0.70 | 92.34±1.04 | 95.18±0.06 | 84.93±0.10 | 93.25±0.57 | **95.79±0.17** | 86.83±0.06 |
| Flights | 95.60±1.73 | 94.03±0.18 | 97.95±0.14 | 98.51±0.01 | 91.86±0.01 | 90.99±0.05 | **98.91±0.01** | 91.85±0.00 |
| Can.Parl | 69.26±0.31 | 70.73±0.72 | 70.88±2.34 | 69.82±2.34 | 54.27±0.12 | 77.04±0.46 | **97.36±0.45** | 95.77±0.18 |
| USLegis | 75.05±1.52 | 68.52±3.16 | 75.99±0.58 | 70.58±0.48 | 54.20±0.08 | 70.74±1.02 | 71.11±0.59 | **92.18±0.23** |
| UNTrade | 64.94±0.31 | 61.47±0.18 | 65.03±1.37 | 65.39±0.12 | 69.02±0.11 | 62.61±0.27 | 66.46±1.29 | **89.19±0.13** |
| UNVote | 63.91±0.81 | 52.21±0.98 | 65.72±2.17 | 52.84±0.10 | 60.64±0.03 | 52.11±0.16 | 55.55±0.42 | **87.29±0.09** |
| Contacts | 95.31±1.33 | 96.28±0.09 | 96.89±0.56 | 90.26±0.28 | 94.63±0.02 | 91.92±0.03 | **98.29±0.01** | 96.10±0.01 |

Table 1: AP for transductive future edge prediction with random negative sampling over five runs. The significantly best result for each benchmark appears in bold font.

over time. In the following experiment, we see that relying on the memory updates solely is not sufficient to achieve reasonable performance in the online future edge prediction setting.

To simulate the online future edge prediction setting, we split the data into 0.1%–1.9%–98% and applied backpropagation at inference time for LDTGN. The other models cannot apply backpropagation at inference time either because they do not have learnable parameters such as Edge-Bank or because they contain an excessive number of learnable parameters as shown in Fig. 5, and hence they cannot withstand the high throughput required for streaming tasks while performing backpropagation. The results of this experiment are reported in Table 3. Here, the partition into a transductive setting and an inductive setting is not required since most of the data are located in the test set and hence this setting resembles the inductive setting already. Furthermore, in Fig. 1b we can see that even when LDTGN performs backpropagation at inference time, it is still more efficient than many other baseline.

## 5.3 ABLATION STUDY

In the previous experiments, all the baselines used a standard batch size of 200 except for EdgeBank, which used a batch size of 1. To maintain fairness, we also used a batch size of 200 for our model. Since, however, LDTGN is decoupled, it can use larger batches without compromising its precision. In Fig. 4 we show the average throughput of our model on the Wikipedia dataset using different batch sizes.

The performance of LDTGN remained consistent across all batch sizes, while its throughput increased with larger batch sizes. The conclusion drawn from this experiment is that temporal networks for dynamic graphs can improve running time performance without any additional cost by simply using the decoupling strategy.

| Dataset | JODIE | TGAT | TGN | CAWN | EdgeBank | GraphMixer | DyGFormer | LDTGN (ours) |
|---|---|---|---|---|---|---|---|---|
| Wikipedia | 94.82±0.20 | 96.22±0.07 | 97.83±0.04 | 98.24±0.03 | 92.90±0.46 | 96.65±0.02 | **98.59±0.03** | 92.61±0.39 |
| Reddit | 96.50±0.13 | 97.09±0.04 | 97.50±0.07 | 98.62±0.01 | 93.18±0.54 | 95.26±0.02 | **98.84±0.02** | 92.95±1.13 |
| MOOC | 79.63±1.92 | 85.50±0.19 | 89.04±1.17 | 81.42±0.24 | 67.22±1.13 | 81.41±0.21 | 86.96±0.43 | **89.10±0.35** |
| LastFM | 81.61±3.82 | 78.63±0.31 | 81.45±4.29 | 89.42±0.07 | 88.04±1.20 | 82.11±0.42 | **94.23±0.09** | 90.82±0.03 |
| Enron | 80.72±1.39 | 67.05±1.51 | 77.94±1.02 | 86.35±0.51 | 94.51±0.37 | 75.88±0.48 | 89.76±0.34 | **95.59±0.35** |
| SocialEvo. | 91.96±0.48 | 91.41±0.16 | 90.77±0.86 | 79.94±0.18 | **96.39±0.21** | 91.86±0.06 | 93.14±0.04 | 95.15±0.32 |
| UCI | 79.86±1.48 | 79.54±0.48 | 88.12±2.05 | 92.73±0.06 | 81.17±0.8 | 91.19±0.42 | **94.54±0.12** | 83.06±0.25 |
| Flights | 94.74±0.37 | 88.73±0.33 | 95.03±0.60 | 97.06±0.02 | 88.60±0.98 | 83.03±0.05 | **97.79±0.02** | 87.06±0.58 |
| Can.Parl. | 53.92±0.94 | 55.18±0.79 | 54.10±0.93 | 55.80±0.69 | 54.36±3.25 | 55.91±0.82 | 87.74±0.71 | **89.95±0.29** |
| USLegis. | 54.93±2.29 | 51.00±3.11 | 58.63±0.37 | 53.17±1.20 | 49.51±0.12 | 50.71±0.76 | 54.28±2.87 | **80.82±0.97** |
| UNTrade | 59.65±0.77 | 61.03±0.18 | 58.31±3.15 | 65.24±0.21 | 64.12±0.97 | 62.17±0.31 | 64.55±0.62 | **89.32±0.56** |
| UNVote | 56.64±0.96 | 52.24±1.46 | 58.85±2.51 | 49.94±0.45 | 59.07±0.45 | 50.68±0.44 | 55.93±0.39 | **86.25±0.31** |
| Contacts | 94.34±1.45 | 95.87±0.11 | 93.82±0.99 | 89.55±0.30 | 94.88±0.25 | 90.59±0.05 | **98.03±0.02** | 96.52±0.22 |

Table 2: AP for inductive future edge prediction with random negative sampling over 5 different runs. The significantly best result for each benchmark appears in bold font.

| Dataset | JODIE | TGAT | TGN | CAWN | EdgeBank | GraphMixer | DyGFormer | LDTGN (ours) |
|---|---|---|---|---|---|---|---|---|
| wikipedia | 49.40±8.44 | 64.89±6.22 | 54.41±13.68 | 90.81±0.88 | **94.21±0.01** | 77.76±1.24 | 75.54±4.43 | **94.22±0.01** |
| reddit | 70.60±3.53 | 62.00±1.04 | 68.30±3.78 | **97.29±0.16** | 94.04±0.01 | 84.11±0.96 | 74.57±4.02 | 94.04±0.01 |
| mooc | 66.02±2.26 | 81.61±0.33 | 76.44±0.67 | 75.82±2.62 | 71.35±0.03 | 75.89±1.45 | 76.59±0.69 | **89.54±0.02** |
| lastFM | 54.90±0.88 | 61.78±1.82 | 55.18±1.97 | 84.39±0.94 | 87.24±0.01 | 70.73±0.81 | 59.83±2.13 | **90.99±0.02** |
| Enron | 49.63±6.48 | 64.98±3.87 | 55.22±3.08 | 75.23±2.23 | 94.53±0.01 | 66.95±2.75 | 66.63±13.61 | **96.02±0.04** |
| SocialEvo | 66.44±0.72 | 79.13±3.19 | 75.44±2.67 | 79.80±1.70 | **94.65±0.01** | 71.44±1.04 | 91.04±0.79 | 93.40±0.02 |
| UCI | 50.26±10.47 | 73.38±4.18 | 79.66±2.86 | 81.60±2.19 | 82.76±0.04 | 58.16±14.26 | 71.97±9.69 | **84.71±0.08** |
| Flights | 76.37±8.90 | 67.97±8.90 | 87.49±1.69 | 51.19±5.64 | 90.23±0.00 | 69.42±10.47 | 82.33±5.97 | **90.23±0.00** |
| Can.Parl | 54.60±9.82 | 56.21±6.15 | 59.48±0.72 | 48.07±3.52 | 55.67±0.06 | 55.28±1.14 | **76.21±9.61** | 80.47±0.06 |
| USLegis | - | 64.19±2.39 | 66.13±0.42 | 50.73±3.74 | 58.97±0.08 | 49.89±3.62 | 50.53±4.15 | **80.43±0.09** |
| UNTrade | 63.42±1.94 | 53.25±4.90 | 62.52±1.39 | 48.87±7.28 | 71.14±0.03 | 52.09±3.54 | 51.04±2.33 | **88.26±0.02** |
| UNVote | 55.35±0.82 | 51.67±2.43 | 56.62±0.72 | 53.30±1.06 | 64.71±0.01 | 50.48±3.61 | 47.73±2.87 | **83.48±0.02** |
| Contacts | 84.25±1.88 | 88.18±1.73 | 88.08±1.29 | 88.30±0.11 | 90.12±0.01 | 87.24±2.27 | 96.09±0.19 | **96.33±0.01** |

Table 3: AP for future edge prediction with random negative sampling over five runs in an online learning setting. The significantly best result for each benchmark appears in bold font.

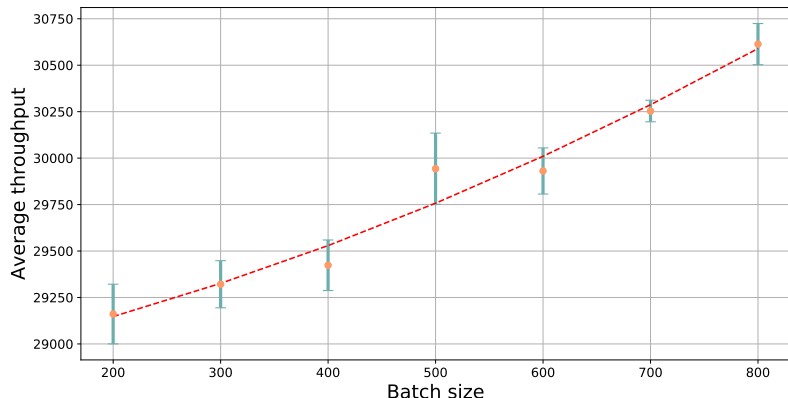

Figure 4: Average throughput of LDTGN on the Wikipedia dataset for future edge prediction using different batch sizes. The means and standard deviations over three consecutive runs are reported as circles and ranges, respectively.

## 6 CONCLUSION

In this work, we presented a decoupling strategy for designing temporal graph networks. Decoupling enables two types of batches to be used – one for the memory module and the other for the prediction module. In this way, temporal graph networks can increase the frequency of the updates while still handling their arrival streams. In addition, we introduced LDTGN – a lightweight model for the future edge prediction task that is extremely efficient in terms of time and memory. We also showed by extensive experiments that LDTGN has outstanding performance for both transductive and inductive tasks while surpassing many other baselines for these tasks. Finally, we observed that the vast majority of the well-known models are not suitable for the scenario of streaming online dynamic link prediction, while our suggested model excels in this setting since it is able to apply backpropagation at inference time.

**Limitations** Although LDTGN achieves great performance on various benchmarks, as reported in Section 5, it does not fully utilize the given data. The model does not use the features of the given inputs. From experiments we conducted, trivial solutions such as appending these features to the results of the memory module do not provide significant benefits. This might be explained by the fact that the timing data is more important in most of the datasets used here. In the datasets where the features might be useful, they have high dimensionality (about 170 features per update). In addition, LDTGN does not utilize the topological structure of the dynamic graph and hence might suffer from the staleness problem (Kazemi et al., 2020). We leave the concepts of effectively and efficiently combining input features and utilizing the full topological structure of the dynamic graph as a future research direction.

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

## A    DATASETS STATISTICS AND DESCRIPTIONS

In our experiments we used the following dynamic graph datasets:

• Wikipedia (Kumar et al., 2019): Wikipedia edit requests log over one month, where the editing users and Wikipedia pages are represented as nodes and the edit requests are modeled as edges. The edges are timestamped and contain LIWC feature vectors (Pennebaker et al., 2001) of the requested text to post.

• Reddit (Kumar et al., 2019): Reddit post requests log over one month where the posting users and subreddits are represented as nodes and the posting requests are modeled as edges.

• MOOC (Kumar et al., 2019): Students' access records to MOOC online courses, where students and content units (e.g., videos, answers, etc.) are described as nodes and the access actions (viewing a video, submitting an answer, etc.) are modeled as edges. The edges are timestamped and have four features describing the action.

• LastFM (Kumar et al., 2019): LastFM listening records over one month, where the LastFM users and the songs are represented as nodes and there is an edge between the users and the songs to which they listened. The edges are timestamped and do not contain any features.

• Enron (Shetty & Adibi, 2004): Email logs of the Enron employees over a period of three years, where the employees are modeled as nodes and a single edge represents an email sent between two employees. The edges are timestamped and do not contain any features.

• Social Evo. (Madan et al., 2011): Documentation of the everyday life of undergraduate students living in dormitories from October 2008 to May 2009. Represented as a mobile phone proximity network where each edge has two features.

• UCI (Panzarasa et al., 2009): Messages logs of the online community of students from the University of California, Irvine, where the students are modeled as nodes and a single edge represents a message sent between two students. The edges are timestamped with a granularity of seconds.

• Flights (Strohmeier et al., 2021): Tracked air traffic during the COVID-19 pandemic, where the airports are modeled as nodes and the edges are the tracked flights between two airports. The edges are timestamped and weighted. The weight of the edges indicates the number of flights between the airports in a day.

• Can. Parl. (Huang et al., 2020): Documented interactions between Canadian members of parliaments from 2006 to 2019, where the members of parliaments are described as nodes, two of which are connected by an edge if they both voted "yes" on a bill. The edges are timestamped and weighted. The weight of the edges indicates the number of times that one member voted "yes" for another member's bill within one year.

• US Legis. (Fowler, 2006): Documented interactions in the US Senate, where legislators are modeled as nodes, two of which are connected by an edge if they co-sponsored a bill. The edges are timestamped and weighted. The weight of the edges indicates the number of times that two members of the US Congress co-sponsored a bill in a given term.

• UN Trade (MacDonald et al., 2015): Documented global food and agriculture trading connections spanning over 30 years, where nations are represented as nodes, two of which are connected by an edge if they have an agriculture import or export relations. The edges are timestamped and weighted. The weight of the edges is the sum of normalized agriculture import or export values between two countries.

• UN Vote (Voeten et al., 2009): Documentation of roll-call votes in the United Nations General Assembly from 1946 to 2020 where nations are represented as nodes, two of which are connected by an edge if they both voted "yes" for an item. The edges are timestamped and weighted. The weight of the edges is the number of times the two countries vote "yes" on a call.

• Contact (Sapiezynski et al., 2019): Physical proximity records documenting around 700 university students over a period of four weeks, where the students are modeled as nodes, two of which are connected by an edge if they each are within close proximity to each other. The edges are timestamped and weighted. The weight of the edges specifies the physical proximity between two students.

The full statistics of the datasets as collected by Yu et al. (2023) are reported in Table 4.

| Dataset | Domain | #Nodes | #Edges | #Edge Features | Bipartite | Duration |
|---|---|---|---|---|---|---|
| Wikipedia | Social | 9,227 | 157,474 | 172 | True | 1 month |
| Reddit | Social | 10,984 | 672,447 | 172 | True | 1 month |
| MOOC | Interaction | 7,144 | 411,749 | 4 | True | 17 months |
| LastFM | Interaction | 1,980 | 1,293,103 | – | True | 1 month |
| Enron | Social | 184 | 125,235 | – | False | 3 years |
| Social Evo. | Proximity | 74 | 2,099,519 | 2 | False | 8 months |
| UCI | Social | 1,899 | 59,835 | – | False | 196 days |
| Flights | Transport | 13,169 | 1,927,145 | 1 | False | 4 months |
| Can. Parl. | Politics | 734 | 74,478 | 1 | False | 14 years |
| US Legis. | Politics | 225 | 60,396 | 1 | False | 12 terms |
| UN Trade | Economics | 255 | 507,497 | 1 | False | 32 years |
| UN Vote | Politics | 201 | 1,035,742 | 1 | False | 72 years |
| Contact | Proximity | 692 | 2,426,279 | 1 | False | 1 month |

Table 4: Datasets statistics.

# B  BASELINES DESCRIPTIONS

In our experiments we used the following temporal graph baselines:

• JODIE (Kumar et al., 2019): JODIE was originally designed to learn the time-evolving nature of temporal bipartite graphs. To do so, it uses two Recurrent Neural Network (RNN) components that learn and update. JODIE also uses a novel projection to create a representation of a future trajectory of each user and item.

•DyRep (Trivedi et al., 2019): DyRep is an RNN-based architecture that utilizes a temporal attention mechanism to exploit the dynamic structure of the graphs.

• TGAT (Xu et al., 2020): TGAT uses a time-encoding function and aggregates neighborhood information using self-attention to compute the embedding for each node.

• TGN (Rossi et al., 2020): TGN is a general architecture for CTDG learning tasks. It uses both a prediction module and a memory module to get relevant and accurate predictions for each input at each moment in time. It does this by aggregating information from the neighborhood of each node and maintain learnable updated memory which is based on RNN, and thus also solves the staleness problem.

• CAWN (Wang et al., 2021b): The CAWN model is based on causal anonymous walks that are generated for each node. The walks are encoded using RNNs and aggregated to achieve the node representation.

• EdgeBank (Poursafaei et al., 2022): EdgeBank is a memorization algorithm that saves any seen update and, given an input, it predicts according to a simple decision rule that can be one of the following: whether the input was seen in the last few iterations (EdgeBank$_{th}$) or in the last few time units (EdgeBank$_{tw}$), or whether the input has already been seen a sufficient number of times (EdgeBank$_{re}$). While EdgeBank can also have a decision rule that is based on infinite memory i.e., predicts positive for any previously seen edge and predicts negative otherwise (EdgeBank$_{inf}$). The algorithm's simplicity allows it to perform extremely fast, making it significantly faster than any other model for dynamic graph learning. In our experiments, we report the best results of EdgeBank among all of its decision rule variations.

• TCL (Wang et al., 2021a): TCL uses an adapted transformer that is able to capture structural and temporal dependency relationships. It also uses a neighborhood encoder that extracts representations of the neighborhood of interacting nodes. The model is optimized by mutual information maximization based on the contrastive learning approach.

• GraphMixer (Cong et al., 2023): GraphMixer uses three components for the task of future edge prediction: a link-encoder that is based on MLP and fixed time-encoding function, a node-encoder that only performs neighborhood mean-pooling and another MLP for edge prediction.

• DyGFormer (Yu et al., 2023): DyGFormer is a transformer-based architecture. To generate an encoding for a given interaction, DyGFormer generates a co-occurrence embedding of the interaction in addition to a neighborhood representation for each interacting node. Then it uses a patching technique on historical representations of the interacting nodes to better capture long-term temporal dependencies. The patches are then sent to a transformer and its outputs are averaged to create the final representation.

## C  ADDITIONAL EXPERIMENTS DETAILS

An additional approach to implementing our model is to split the prediction function into two linear classifiers instead of having just one classifier. In this setting, we used the original classifier in Eq. (8) where the edge to predict was previously seen and Eq. (15) where the edge to predict was not previously seen but the two interacting nodes are both not new.

$$p = (t - t_i)w_4 + (t - t_j)w_5 + c \tag{15}$$

In our experiments, we tuned our model between using one classifier and using two classifiers and reported the best results. Furthermore, we used stochastic gradient descent as an optimizer with a learning rate of 0.0001, with mean squared error as the objective function.

In the setting in which we trained our model, it is also possible to perform forward-backward propagation not only on the current batch but also to include a few of the previous batches to achieve possibly better performance. To do so, the states of each previously trained batch must be saved until the full batch finishes participating in the training, since the memory gets updated upon receiving a new batch and consequently the states of the previous batch are overridden. In our experiments, we did not use previous batches for training.

All the experiments were performed on RTX2080ti and Intel(R)Xeon(R)Silver4108.

# D    ADDITIONAL RESULTS

In Fig. 5 we present the average number of learnable parameters each model used in our experiments. We can see that our model needs a significantly lower number of learnable parameters compared to the other baselines, which makes it practical to be used in the task of online future edge prediction. EdgeBank does not contain learnable parameters at all and, therefore, it applies the same prediction strategy throughout time even in an online scenario.

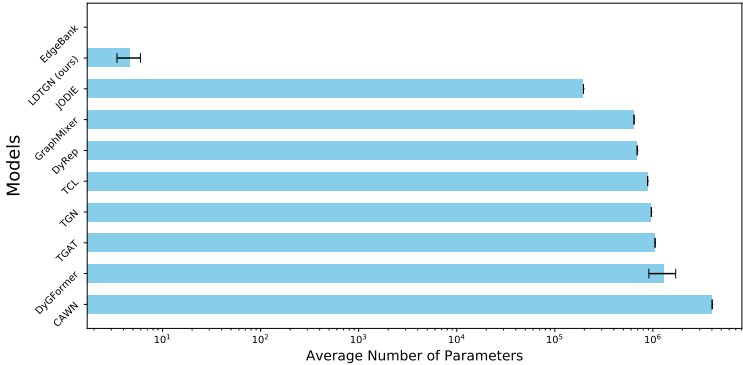

Figure 5:   Average number of learnable parameters used by each baseline in the experiments.

In Fig. 6 we show the average throughput (processed edges per second) of each baseline used in our standard transductive and inductive experiments.

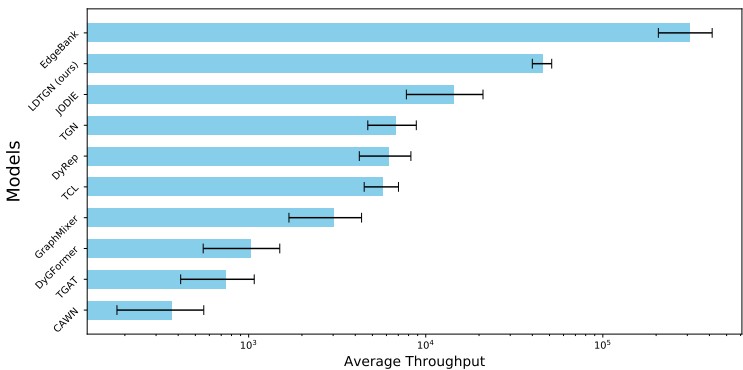

Figure 6: Average throughput (processed edges per second) of each baseline in the experiments.

In Table 5, Table 6 and Table 7 we report the full results of the transductive, inductive and online experiments, respectively.

| Dataset | JODIE | DyRep | TGAT | TGN | CAWN | EdgeBank | TCL | GraphMixer | DyGFormer | LDTGN (ours) |
|---|---|---|---|---|---|---|---|---|---|---|
| wikipedia | 96.50±0.14 | 94.86±0.06 | 96.94±0.06 | 98.45±0.06 | 98.76±0.03 | 94.41±0.02 | 96.47±0.16 | 97.25±0.03 | **99.03±0.02** | 94.43±0.03 |
| reddit | 98.31±0.14 | 98.22±0.04 | 98.52±0.02 | 98.63±0.06 | 99.11±0.01 | 95.78±0.02 | 97.53±0.02 | 97.31±0.01 | **99.22±0.01** | 95.80±0.01 |
| mooc | 80.23±2.44 | 81.97±0.49 | 85.84±0.15 | 89.15±1.60 | 80.15±0.25 | 71.46±0.08 | 82.38±0.24 | 82.78±0.15 | 87.52±0.49 | **94.08±0.04** |
| lastFM | 70.85±2.13 | 71.92±2.21 | 73.42±0.21 | 77.07±3.97 | 86.99±0.06 | 86.13±0.03 | 67.27±2.16 | 75.61±0.24 | **93.00±0.12** | 91.28±0.03 |
| Enron | 84.77±0.30 | 82.38±3.36 | 71.12±0.97 | 86.53±1.11 | 89.56±0.09 | 91.02±0.21 | 79.70±0.71 | 82.25±0.16 | 92.47±0.12 | **95.67±0.06** |
| SocialEvo | 89.89±0.55 | 88.87±0.30 | 93.16±0.17 | 93.57±0.17 | 84.96±0.09 | 94.64±0.02 | 93.13±0.16 | 93.37±0.07 | **94.73±0.01** | 93.12±0.06 |
| UCI | 89.43±1.09 | 65.14±2.30 | 79.63±0.70 | 92.34±1.04 | 95.18±0.06 | 84.93±0.10 | 89.57±1.63 | 93.25±0.57 | **95.79±0.17** | 86.83±0.06 |
| Flights | 95.60±1.73 | 95.29±0.72 | 94.03±0.18 | 97.95±0.14 | 98.51±0.01 | 91.86±0.01 | 91.23±0.02 | 90.99±0.05 | **98.91±0.01** | 91.85±0.00 |
| Can.Parl | 69.26±0.31 | 66.54±2.76 | 70.73±0.72 | 70.88±2.34 | 69.82±2.34 | 54.27±0.12 | 68.67±2.67 | 77.04±0.46 | **97.36±0.45** | 95.77±0.18 |
| USLegis | 75.05±1.52 | 75.34±0.39 | 68.52±3.16 | 75.99±0.58 | 70.58±0.48 | 54.20±0.08 | 69.59±0.48 | 70.74±1.02 | 71.11±0.59 | **92.18±0.23** |
| UNTrade | 64.94±0.31 | 63.21±0.93 | 61.47±0.18 | 65.03±1.37 | 65.39±0.12 | 69.02±0.11 | 62.21±0.03 | 62.61±0.27 | 66.46±1.29 | **89.19±0.13** |
| UNVote | 63.91±0.81 | 62.81±0.80 | 52.21±0.98 | 65.72±2.17 | 52.84±0.10 | 60.64±0.03 | 51.90±0.30 | 52.11±0.16 | 55.55±0.42 | **87.29±0.09** |
| Contacts | 95.31±1.33 | 95.98±0.15 | 96.28±0.09 | 96.89±0.56 | 90.26±0.28 | 94.63±0.02 | 92.44±0.12 | 91.92±0.03 | **98.29±0.01** | 96.10±0.01 |

Table 5: AP for transductive future edge prediction with random negative sampling over five runs. The significantly best result for each benchmark appears in bold font.

| Dataset | JODIE | DyRep | TGAT | TGN | CAWN | EdgeBank | TCL | GraphMixer | DyGFormer | LDTGN (ours) |
|---|---|---|---|---|---|---|---|---|---|---|
| Wikipedia | 94.82±0.20 | 92.43±0.37 | 96.22±0.07 | 97.83±0.04 | 98.24±0.03 | 92.90±0.46 | 96.22±0.17 | 96.65±0.02 | **98.59±0.03** | 92.61±0.39 |
| Reddit | 96.50±0.13 | 96.09±0.11 | 97.09±0.04 | 97.50±0.07 | 98.62±0.01 | 93.18±0.54 | 94.09±0.07 | 95.26±0.02 | **98.84±0.02** | 92.95±1.13 |
| MOOC | 79.63±1.92 | 81.07±0.44 | 85.50±0.19 | 89.04±1.17 | 81.42±0.24 | 67.22±1.13 | 80.60±0.22 | 81.41±0.21 | 86.96±0.43 | **89.10±0.35** |
| LastFM | 81.61±3.82 | 83.02±1.48 | 78.63±0.31 | 81.45±4.29 | 89.42±0.07 | 88.04±1.20 | 73.53±1.66 | 82.11±0.42 | **94.23±0.09** | 90.82±0.03 |
| Enron | 80.72±1.39 | 74.55±3.95 | 67.05±1.51 | 77.94±1.02 | 86.35±0.51 | 94.51±0.37 | 76.14±0.79 | 75.88±0.48 | 89.76±0.34 | **95.59±0.35** |
| SocialEvo. | 91.96±0.48 | 90.04±0.47 | 91.41±0.16 | 90.77±0.86 | 79.94±0.18 | **96.39±0.21** | 91.55±0.09 | 91.86±0.06 | 93.14±0.04 | 95.15±0.32 |
| UCI | 79.86±1.48 | 57.48±1.87 | 79.54±0.48 | 88.12±2.05 | 92.73±0.06 | 81.17±0.80 | 87.36±2.03 | 91.19±0.42 | **94.54±0.12** | 83.06±0.25 |
| Flights | 94.74±0.37 | 92.88±0.73 | 88.73±0.33 | 95.03±0.60 | 97.06±0.02 | 88.60±0.98 | 83.41±0.07 | 83.03±0.05 | **97.79±0.02** | 87.06±0.58 |
| Can.Parl. | 53.92±0.94 | 54.02±0.76 | 55.18±0.79 | 54.10±0.93 | 55.80±0.69 | 54.36±3.25 | 54.30±0.66 | 55.91±0.82 | 87.74±0.71 | **89.95±0.29** |
| USLegis. | 54.93±2.29 | 57.28±0.71 | 51.00±3.11 | 58.63±0.37 | 53.17±1.20 | 49.51±0.12 | 52.59±0.97 | 50.71±0.76 | 54.28±2.87 | **80.82±0.97** |
| UNTrade | 59.65±0.77 | 57.02±0.69 | 61.03±0.18 | 58.31±3.15 | 65.24±0.21 | 64.12±0.97 | 62.21±0.12 | 62.17±0.31 | 64.55±0.62 | **89.32±0.56** |
| UNVote | 56.64±0.96 | 54.62±2.22 | 52.24±1.46 | 58.85±2.51 | 49.94±0.45 | 59.07±0.45 | 51.60±0.97 | 50.68±0.44 | 55.93±0.39 | **86.25±0.31** |
| Contacts | 94.34±1.45 | 92.18±0.41 | 95.87±0.11 | 93.82±0.99 | 89.55±0.30 | 94.88±0.25 | 91.11±0.12 | 90.59±0.05 | **98.03±0.02** | 96.52±0.22 |

Table 6: AP for inductive future edge prediction with random negative sampling over five different runs. The significantly best result for each benchmark appears in bold font.

| Dataset | JODIE | DyRep | TGAT | TGN | CAWN | EdgeBank | TCL | GraphMixer | DyGFormer | LDTGN (ours) |
|---|---|---|---|---|---|---|---|---|---|---|
| wikipedia | 49.40±8.44 | 52.11±3.34 | 64.89±6.22 | 54.41±13.68 | 90.81±0.88 | **94.21±0.01** | 64.57±4.40 | 77.76±1.24 | 75.54±4.43 | **94.22±0.01** |
| reddit | 70.60±3.53 | 58.89±2.68 | 62.00±1.04 | 68.30±3.78 | **97.29±0.16** | 94.04±0.01 | 84.11±3.03 | 84.57±4.02 | 74.57±4.02 | 94.04±0.01 |
| mooc | 66.02±2.26 | 49.25±8.57 | 81.61±0.33 | 76.44±0.67 | 75.82±2.62 | 71.35±0.03 | 76.53±1.12 | 75.89±1.45 | 76.59±0.69 | **89.54±0.02** |
| lastFM | 54.90±0.88 | 52.51±1.33 | 61.78±1.82 | 55.18±1.97 | 84.39±0.94 | 87.24±0.01 | 55.44±2.44 | 70.73±0.81 | 59.83±2.13 | **90.99±0.02** |
| Enron | 49.63±6.48 | 51.58±7.22 | 64.98±3.87 | 55.22±3.08 | 75.23±2.23 | 94.53±0.01 | 61.60±3.89 | 66.95±2.75 | 66.63±13.61 | **96.02±0.04** |
| SocialEvo | 66.44±0.72 | 53.32±1.40 | 79.13±3.19 | 75.44±2.67 | 79.80±1.70 | **94.65±0.01** | 74.06±1.39 | 71.44±1.04 | 91.04±0.79 | 93.40±0.02 |
| UCI | 50.26±10.47 | 51.31±9.86 | 73.38±4.18 | 79.66±2.86 | 81.60±2.19 | 82.76±0.04 | 51.05±4.86 | 58.16±14.26 | 71.97±9.69 | **84.71±0.08** |
| Flights | 76.37±8.90 | 76.92±3.59 | 67.97±8.90 | 87.49±1.69 | 51.19±5.64 | **90.23±0.00** | 56.02±8.53 | 69.42±10.47 | 82.33±5.97 | **90.23±0.00** |
| Can.Parl | 54.60±9.82 | 58.03±2.12 | 56.21±6.15 | 59.48±0.72 | 48.07±3.52 | 55.67±0.06 | 51.86±4.03 | 55.28±1.14 | **76.21±9.61** | 80.47±0.06 |
| USLegis | - | 64.19±2.39 | 52.76±1.30 | 66.13±0.42 | 50.73±3.74 | 58.97±0.08 | 55.45±3.94 | 49.89±3.62 | 50.53±4.15 | **80.43±0.09** |
| UNTrade | 63.42±1.94 | 63.09±0.24 | 53.25±4.90 | 62.52±1.39 | 48.87±7.28 | 71.14±0.03 | 46.04±3.91 | 52.09±3.54 | 51.04±2.33 | **88.26±0.02** |
| UNVote | 55.35±0.82 | 56.03±0.42 | 51.67±2.43 | 56.62±0.72 | 53.30±1.06 | 64.71±0.01 | 50.77±2.51 | 50.48±3.61 | 47.73±2.87 | **83.48±0.02** |
| Contacts | 84.25±1.88 | 54.15±2.42 | 88.18±1.73 | 88.08±1.29 | 88.30±0.11 | 90.12±0.01 | 85.25±1.17 | 87.24±2.27 | 96.09±0.19 | **96.33±0.01** |

Table 7: AP for future edge prediction with random negative sampling over five runs in an online learning setting. The significantly best result for each benchmark appears in bold font.

