# OpenReview forum: "Leveraging Temporal Graph Networks Using Module Decoupling"
_ICLR.cc/2024/Conference — ICLR 2024 Conference Withdrawn Submission_

### Official Review · Reviewer_VFFL · 2023-10-15

**Soundness:** 3 good
**Presentation:** 1 poor
**Contribution:** 2 fair
**Rating:** 3
**Confidence:** 4

**Summary:**

This work studies the problem of future edge prediction (inductive or transductive) in the streaming graph setting. To achieve high throughput, the work decouples the memory and prediction modules of Temporal Graph Network (TGN, Rossi et al., 2020), where the memory module uses smaller batch sizes for frequent updates and the prediction module uses larger batch sizes for efficiency. To improve prediction power, the work parameterizes the EdgeBank (Poursafaei et al., 2022) model to allow it to learn.

**Strengths:**

A lot of experiments were conducted comparing with many alternative models.

**Weaknesses:**

The writing is substandard. For example, the model description on Page 6 is very unclear. For example, what are the subscripts 2 in Eq(10) and 1 in Eq(11)?
Section 5.1 refers to Table 2 which is 1 full page later and Fig. 6 which is on Page 17! Page 8 refers to Fig.1b which is on Page 2...

Also, the model is very simple and incremental to TGN and EdgeBank.

On Page 8, it is claimed that "The other models cannot apply back- propagation at inference time" so you are comparing your online trained model with the other untrained ones, so the performance is not fair. Even so, your performance is not always the best. In particular, in Table 1, DyGFormer is better on more experiments than your method, and the first one has a large gap 99.03 vs 94.43.

Finally, you discussed the model limitations like not considering features and topological structure, which seem very essential to me. My overall concern is that you are using a very simple model so that online training is possible, but the accuracy is far from more advanced temporal GCN models.

**Questions:**

N/A

---

### Official Review · Reviewer_CHuW · 2023-10-31

**Soundness:** 3 good
**Presentation:** 2 fair
**Contribution:** 2 fair
**Rating:** 3
**Confidence:** 5

**Summary:**

This paper proposes a decoupling strategy for designing temporal graph networks, which enables the models to update frequently while using batches. Specifically, decoupling enables two types of batches to be used, one for the memory module and the other for the prediction module. In this way, temporal graph networks can increase the frequency of the updates while still handling their arrival streams. In addition, the author presents a lightweight model named Lightweight Decoupled Temporal Graph Network (LDTGN) for the future edge prediction task, which is exceptionally efficient in terms of time and memory.

**Strengths:**

1. It is attractive and intuitive to modify the Temporal Graph Network with a decoupling strategy. Decoupling enables the batches for the memory and the prediction module to be different, which can increase the frequency of the updates while keeping throughput.
2. The paper is well-organized with a logical flow.

**Weaknesses:**

1. The problem has not yet been well motivated. The authors point out the problem of missing updates but do not tell what consequences the problem brings to the embedding models. Give a running example in the Introduction section would help understand the problem better. It is unclear to the audience how significant the problem is and why existing approaches fail to solve it properly. The authors mentioned some related work in the paper but did not point out their limitations. Therefore, it is very difficult to justify the contributions of this work.
2. The proposed method is not well justified. Yes, the proposed method is a way to solve the problem but it is questionable whether it is one of the best ways for solving the problem. No technical challenges are identified. What are the intuitions behind the proposed solutions?
3. The experiments are not solid. The authors indicated that "missing updates are less likely to occur when using smaller batches since, in these circumstances, the models update more frequently." As a result, the batch size is a very important parameter of the models. However, the impact of batch size on the precision of edge prediction has not yet been tested in the experiments. Simply testing its impact on the time throughput cannot answer this question. In addition, the memory cost of the proposed algorithms should be tested.

**Questions:**

1. What is the memory throughput of the proposed methods regarding the varying batch size?
2. Is the proposed strategy applicable to any continuous-time dynamic graph embedding algorithms? If yes, why and how?

---

### Official Review · Reviewer_VY6D · 2023-11-05

**Soundness:** 3 good
**Presentation:** 2 fair
**Contribution:** 2 fair
**Rating:** 3
**Confidence:** 5

**Summary:**

The paper focuses on the link prediction problem in streaming settings for continuous-time dynamic graphs. The authors propose a module decoupling strategy, which separates the memory and prediction modules of the TGN model and employs different batch sizes. The memory module utilizes a smaller batch size for more frequent updates, while the prediction module employs a larger batch size to enhance efficiency.

**Strengths:**

S1. Designing GNN models for continuous-time dynamic graphs with high streaming rates is an interesting direction.

S2. The idea of decoupling memory and prediction modules is simple and intuitive.

**Weaknesses:**

W1. I am not fully convinced by the motivation of his problem. Firstly, the author has not clearly articulated why the issue of missing updates is significant and why existing methods struggle to address it. Secondly, LDTGN is a parameterized version of the baseline EdgeBank, and this enhancement utilizes the modular design approach of TGN. However, the intuition behind proposing this solution has not been elaborately explained.

W2. The features of nodes and edges have not been considered, and it seems that the approach is only capable of handling scenarios involving the addition of edges.

W3. The datasets used in the paper are relatively small. Since running time is a major concern of the paper, I would suggest the authors use larger datasets to demonstrate the effectiveness of LDTGN, and include baseline [1].

W4. The comparison might be unfair. I would suggest the authors apply a similar online learning strategy to baselines, ensuring a more equitable comparison. Additionally, the authors can integrate accuracy and latency metrics to demonstrate the advantages of LDTGN.

[1] Zheng Y, Wei Z, Liu J. Decoupled graph neural networks for large dynamic graphs. Proc. VLDB Endow., 2023, 16(9): 2239–2247.

**Questions:**

In addition to the points mentioned in W1-W4, the authors should also consider revising the presentation of the paper. For example, $w_1$、$w_2$、$w_3$ and $b$ in Equation 8 should be clearly defined.